# Predicting hospital readmission risk: A prospective observational study to compare primary care providers' assessments with the LACE readmission risk index

Sakina Walji[1,2]*, Warren McIsaac[1,2], Rahim Moineddin[2,3], Sumeet Kalia[2,4,5], Michelle Levy[1], Karen Tu[2,5,6], Chaim M. Bell[1,7,8]

1 Ray D. Wolfe Department of Family Medicine, Mount Sinai Hospital, Toronto, Canada, 2 Department of Family and Community Medicine, University of Toronto, Toronto, Ontario, Canada, 3 Primary Care and Health Systems Research Program, ICES, Toronto, Ontario, Canada, 4 University of Toronto Practice-Based Research Network (UTOPIAN), Toronto, Ontario, Canada, 5 North York General Hospital, North York, Ontario, Canada, 6 Institute of Health Policy, Management and Evaluation, University of Toronto, Toronto, Ontario, Canada, 7 Department of Medicine, University of Toronto, Toronto, Ontario, Canada, 8 Department of Health Policy, Management and Evaluation, University of Toronto, Toronto, Ontario, Canada

* sakina.walji@sinaihealth.ca

**Data Availability Statement:** All data underlying the findings described in the manuscript are not fully available/without restriction. The data used

## Abstract

### Purpose

This study aims to determine if the primary care provider (PCP) assessment of readmission risk is comparable to the validated LACE tool at predicting readmission to hospital.

### Methods

A prospective observational study of recently discharged adult patients clustered by PCPs in the primary care setting. Physician readmission risk assessment was determined via a questionnaire after the PCP reviewed the hospital discharge summary. LACE scores were calculated using administrative data and the discharge summary. The sensitivity and specificity of the physician assessment and the LACE tool in predicting readmission risk, agreement between the 2 assessments and the area under receiver operating characteristic (AUROC) curves were calculated.

### Results

217 patient readmission encounters were included in this study from September 2017 till June 2018. The rate of readmission within 30 days was 14.7%, and 217 discharge summaries were used for analysis. The weighted kappa coefficient was 0.41 (95% CI: 0.30–0.51) demonstrating a moderate level of agreement. Sensitivity of physician assessment was 0.31 (95% CI: 0.22–0.40) and specificity was 0.80 (95% CI: 0.77–0.83). The sensitivity of the LACE assessment was 0.42 (95% CI: 0.25–0.59) and specificity was 0.79 (95% CI: 0.73–0.85). The AUROC for the LACE readmission risk was 0.65 (95% C.I. 0.55–0.76)

within this study was obtained through the Institute for Clinical and Evaluative Sciences (ICES), and as such there are legal restrictions to sharing the dataset. As well, given that the data includes patient health information, there are ethical restrictions in regard to the sharing of data, whereby the lead study institution, Sinai Health System, would require a data transfer agreement to be approved and in place prior to transferring or sharing the data. Requests for access to the data should be directed to Dr. Sakina Walji, at sakina.walji@sinaihealth.ca, and Stefana Jovanovska, at stefana.jovanovska@ices.on.ca.

**Funding:** This study received grant funding from the University of Toronto's UTOPIAN Ideas to Proposal program. The funder provided support to cover the cost of conducting the study but did not have any additional role in the study design, data collection and analysis, decision to publish or preparation of the manuscript.

**Competing interests:** Chaim Bell, a co-author on this project, is a medical consultant to the Ontario Ministry of Health, however the Ministry of Health had no role in this study. This commercial affiliation does not alter our adherence to PLOS ONE policies on sharing data and materials. There are no additional competing interests to declare.

**Abbreviations:** PCP, Primary care provider; OHIP, Ontario Health Insurance Plan; AUROC, area under receiver operating characteristic curves; DAS, Data and Analytic Services; SPOR, Strategy for Patient-Oriented Research; AUC, area under the curve.

demonstrating modest predictive power and was 0.57 (95% C.I. 0.46–0.68) for physician assessment, demonstrating low predictive power.

## Conclusion

The LACE index shows moderate discriminatory power in identifying high-risk patients for readmission when compared to the PCP's assessment. If this score can be provided to the PCP, it may help identify patients who requires more intensive follow-up after discharge.

## Introduction

Readmission after hospitalization is common and costly [1, 2]. It affects almost one in ten of all hospitalizations and up to one-third may be avoidable [1, 3, 4]. Many jurisdictions have identified 30-day hospital readmissions as a key quality indicator [5]. Similarly, strategies to reduce rates of readmissions have become a priority in many countries [6, 7].

Several studies provide support for the effectiveness of early follow-up post-hospital discharge in the primary care setting to reduce readmission [8–11]. Patients aged 65 and over who do not have physician follow-up within 30 days of discharge are three times more likely to be re-admitted [8]. Follow-up post-discharge was associated with a 19% lower chance of readmission for patients with congestive heart failure [9], and a 15% reduction in readmission to hospital after high-risk surgery associated with complications [10]. Follow-up within 7 days post-discharge from hospital in the primary care setting has been used in primary care quality frameworks [11].

A number of readmission risk assessment tools have been developed to identify high-risk patients who may benefit from early post-discharge follow-up [12–15]. The LACE index is a validated score that assesses the risk of death or unplanned readmission after discharge from hospital [12]. This tool has been suggested to help identify patients needing post-discharge interventions [16]. However, some information needed to calculate the score, such as emergency room visits in the previous 6 months, may not be readily available to primary care providers (PCPs) at the time an assessment regarding the need for early post-discharge follow-up is being made.

Primary care providers who possess detailed clinical and social knowledge of patients in the community gained over time may be able to predict which patients are at higher risk for hospital readmission. This study aims to determine if the primary care provider assessment of readmission risk is comparable to the validated LACE tool at predicting readmission to hospital within 30 days, for patients who have been discharged from hospital within the last 14 days.

## Methods

### Study setting and design

This study took place in Toronto, Canada between September 2017 and June 2018. Three family practice clinics of varying size participated. The sites included 31 physician faculty and a total of approximately 27,500 rostered patients in the sampling frame of the study. This study received ethics approval from the Mount Sinai Hospital Research Ethics Board (MSH REB #17-0173-E). Written consent was obtained for all participating clinicians.

This was a prospective observational study of recently discharged adult patients clustered by PCPs to observe agreement between PCP risk assessment of readmission compared to the

LACE tool against a criterion standard of observed hospital readmission within 30 days in any hospital in Ontario, Canada.

## Determination of PCP and LACE risk estimates for readmission

To determine PCP estimate of the risk for an individual patient of being readmitted, hospital discharge summaries from the patient's electronic medical record or faxed to the clinic from the hospital were identified from the period of September 2017 to June 2018. Only discharge summaries for adults 18 years of age or older cared for by a consenting study physician, and received within 14 days of hospital discharge, were eligible.

A survey was attached to each hospital discharge summary that asked the PCP to rate their risk of readmission for that patient within 30 days as low, moderate or high, as well as the factors that led him/her to that decision (S1 File).

The LACE tool provides a risk estimate of readmission based on clinical and administrative data elements to provide a risk score from 0–19. The variables comprising the score include length of stay ("L"); acuity of the admission ("A"); comorbidity of the patient (measured with the Charlson comorbidity index score) ("C") [17]; and emergency department use (measured as the number of visits in the six months before admission) ("E") [12]. The Charlson comorbidity index is determined by assigning a score for each comorbid condition depending on the risk of dying associated with each condition. In the original study, the LACE score for each patient was classified into (i) low risk (with LACE score 0–8) corresponding to < 10% risk of readmission, (ii) medium risk (with LACE score 9–13) corresponding to 10–20% risk of readmission and (iii) high risk (with LACE score 14–19) corresponding to >20% risk of readmission [12].

Ontario has universal health care coverage for visits to doctors in emergency departments, clinics, and for hospital admissions under the Ontario Health Insurance Plan (OHIP). These visits are captured using a unique identifier for each eligible resident of the province and all analyses are considered population based. This administrative data is available for research upon request through ICES (www.ices.on.ca, n.d.). Discharge summaries were linked to the administrative data through a combination of patient identifiers which included hospital number, patient name, hospital of admission and date of admission. Administrative data was also used to determine the number of emergency department visits in the prior 6 months, as well as whether the patient was readmitted within 30 days. This was then linked to the data from the hospital discharge summary and physician risk estimate. Anonymized data was sent back to the research team through a secure portal. Data on length of stay, acuity and co-morbidities was retrieved manually from the discharge summary.

## Statistical analysis

Agreement between the LACE and physician risk assessment was quantified using the weighted kappa statistic to account for differences on ordinal scale of predicting 30-days readmission (low, medium, high). The weighted kappa statistic can be interpreted as indicating slight agreement (0.01–0.20), fair agreement (0.21–0.40), moderate agreement (0.41–0.60), substantial agreement (0.61–0.80), and almost perfect agreement (0.80–0.99) [18]. The association between the readmission rate derived from physician risk assessment and LACE tool was assessed using two-sided Cochrane-Armitage trend test.

Sensitivity and specificity for predicting a high risk for hospital readmission was estimated by collapsing the "low risk" and "medium risk" categories into a single group for the PCP assessment, as indicated by the treating PCP. This was done to be consistent with the LACE categorization of high-risk readmissions as per the original paper. We felt it would be most

prudent to identify and closely follow-up those patients deemed to be high-risk for readmission. These diagnostic measures were estimated for the LACE tool and physician assessment.

The area under receiver operating characteristic curves (AUROC) was used to estimate the predictive power of 30-days readmission for LACE score and physician assessment. An AUROC value of 0.5 suggests no discriminatory predictive power, 0.7 to 0.8 as acceptable discriminatory predictive power and greater than 0.8 as excellent discriminatory predictive power [19]. We compared the AUROC of LACE tool and physician assessment tool using the non-parametric approach, as further described by DeLong, Delong and Clarke-Pearson [20].

Multiple logistic regression models were fitted to describe the relationship for LACE score and physician assessment with respect to the 30-days readmission, while adjusting for patient's age and gender [21]. We conducted complete case analysis when fitting the multivariable logistic regressions.

All analyses were conducted using SAS v9.4.

This study contracted ICES Data and Analytic Services (DAS) and used de-identified data from the ICES Data Repository, which is managed by ICES with support from its funders and partners: Canada's Strategy for Patient-Oriented Research (SPOR), the Ontario SPOR Support Unit, the Canadian Institute of Health Research and the Government of Ontario. The opinions, results and conclusions reported are those of the authors. No endorsement by ICES or any of its funders or partners is intended or should be inferred.

## Results

Twenty-one of 31 eligible physicians (67.7%) agreed to participate in the study. A total of 257 discharge summaries were collected across the 3 sites during the study period. Eight summaries were excluded as they included pediatric or obstetric cases. Physicians completed survey questionnaires for 238 (96.6%), of which 21 were excluded as the provider was aware of the readmission prior to completing the survey leaving 217 on which analysis was completed. The LACE score was able to be calculated for 208 patients and physician assessments were available for 202 patients (Fig 1).

The overall rate of readmission within 30 days was 32/217 (14.7%). There was no statistically significant difference in proportion of readmission when comparing patients less than 64 years of age with more than 65 years of age, males with females or low-income group with high income group (Table 1).

Both the LACE tool and physician assessment identified a higher proportion of older patients (age 65+) than younger patients, a higher proportion of male than female patients and a higher proportion of low income group than high income group to be at high risk of readmission.

Patients with at least one emergency visit in the previous 6 months had 18.6% readmission rate while patients with no emergency visit had 9.1% readmission rate (p = 0.05) (Table 2).

The LACE tool categorised 97/208 (46.6%) as low risk, 60/208 (28.8%) as medium risk and 51/208 (24.5%) as high risk. Physicians estimated the risk of readmission low for 101/202 (50.0%), medium for 58/202 (28.7%) and high for 43/202 (21.3%).

### Agreement between physician risk assessment and LACE instrument estimates

The weighted kappa coefficient was 0.41 (95% CI: 0.30–0.51) demonstrating the presence of statistically significant agreement between the LACE readmission risk tool and physician assessment demonstrating a statistically significant moderate level of agreement.

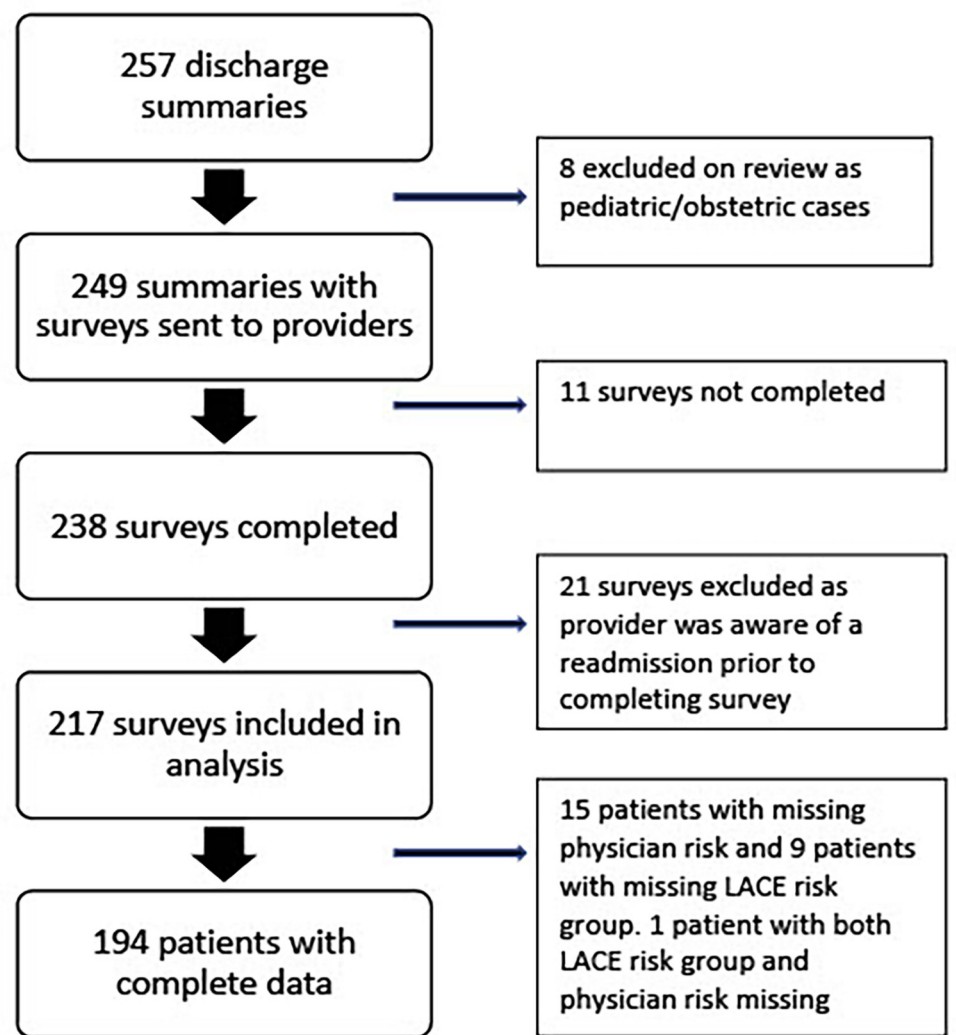

**Fig 1. Number of patients included in the study.**

**Table 1. Characteristics of the patients involved in the study.**

| | Readmission within 30 days after hospital discharge (with respect to hospital discharge) | | |
|---|---|---|---|
| | **No** | **Yes** | **Total** |
| **Age group (year)** | | | |
| 18–64 | 80 (82.5%) | 17 (17.5%) | N = 97 |
| ≥ 65 | 90 (85.7%) | 15 (14.3%) | N = 105 |
| **Gender** | | | |
| Female | 92 (85.2%) | 16 (14.8%) | N = 108 |
| Male | 78 (83%) | 16 (17%) | N = 94 |
| **Income group** * | | | |
| High | 115 (83.3%) | 23 (16.7%) | N = 138 |
| Low | 55 (85.9%) | 9 (14.1%) | N = 64 |

[1]categorized using income quintiles where [1, 2, 3] is low income and [4, 5] is high income.

**Table 2. Risk of readmission associated with increased co-morbidity, acuity of admission and previous number of emergency department visits.**

| Category | Readmission rate |
|---|---|
| Two or more co-morbidities vs. 1 or less | 18.2% vs. 11.2% (p = 0.16) |
| Admitted for 2 or more days vs. 1 or less | 15.2% vs. 14.0% (p = 0.84) |
| Acute vs. elective admission | 17.1% vs. 7.8% (p = 0.11) |
| **One or more vs. no ED visits in the previous 6 months** | **18.6% vs. 9.1% (p = 0.05)** |

When LACE score deemed patient to be low risk of readmission, there was 76% agreement with physician assessment. In contrast, when LACE score deemed patient to be high risk, there was 43% agreement with physician assessment. Overall, there was 57.8% agreement (i.e. (71 +20+21)/194) between physician risk assessment and LACE risk groups (Table 3).

The odds of being readmitted within 30 days increased by 1.16 with every one point increase in LACE score (95% CI: 1.04–1.28; p-value = 0.004). The AUROC estimated for the LACE readmission risk estimate was 0.65 (95% C.I. 0.55–0.76) demonstrating modest predictive power. The AUROC estimate for the physician risk for readmission suggested a lower level of discrimination with a value of 0.57 (95% C.I. 0.46–0.68) (Fig 2). However, the AUROC of LACE readmission was not statistically significant at the nominal level of 5% to the AUROC of physician assessment (Chi-square = 3.11, p = 0.08).

When adjusting for age range and gender, the odds of 30-days readmission among patients labelled in high risk group by LACE score was 3.61 times higher than the odds of readmission among patients labelled in low risk group by LACE score (95% CI: 1.37–9.55; p-value = 0.009).

## Accuracy of physician and LACE estimates of risk of readmission

There was a 9% rate of readmission within 30 days when LACE score was low risk, 15% risk of readmission when LACE score was medium risk and 25% rate of readmission when LACE score was high risk, indicating statistically significant trend in the increasing rate of re-admission with respect to LACE score (P-value = 0.009) (Table 4). Similarly with physician assessment, there was a 12% readmission rate within 30 days when physician assessment was deemed to be low risk, 14% risk of readmission when physician assessment was medium risk and 21% risk of readmission when physician assessment was high risk, indicating statistically insignificant trend in the increasing rate of re-admission with respect to physician assessment (P-value = 0.18) (Table 4).

The diagnostic estimates for the physician risk assessment were: (1) sensitivity = 0.31 (95% CI: 0.22–0.40) and (2) specificity = 0.80 (95% CI: 0.77–0.83). The diagnostic estimates for LACE group were: (1) sensitivity = 0.42 (95% CI: 0.25–0.59) and (2) specificity = 0.79 (95% CI: 0.73–0.85).

**Table 3. Agreement between physician risk assessment and LACE tool.**

| LACE risk group[1] | Physician Risk Assessment | | | |
|---|---|---|---|---|
| | Low | Medium | High | Total |
| Low | **71** | 16 | 7 | 94 |
| Medium | 19 | **20** | 12 | 51 |
| High | 9 | 19 | **21** | 49 |
| **Total** | 99 | 55 | 40 | **194** |

[1]categorized using the LACE score (0–9 = low risk; 10–13 = medium risk; 14–19 = high risk).

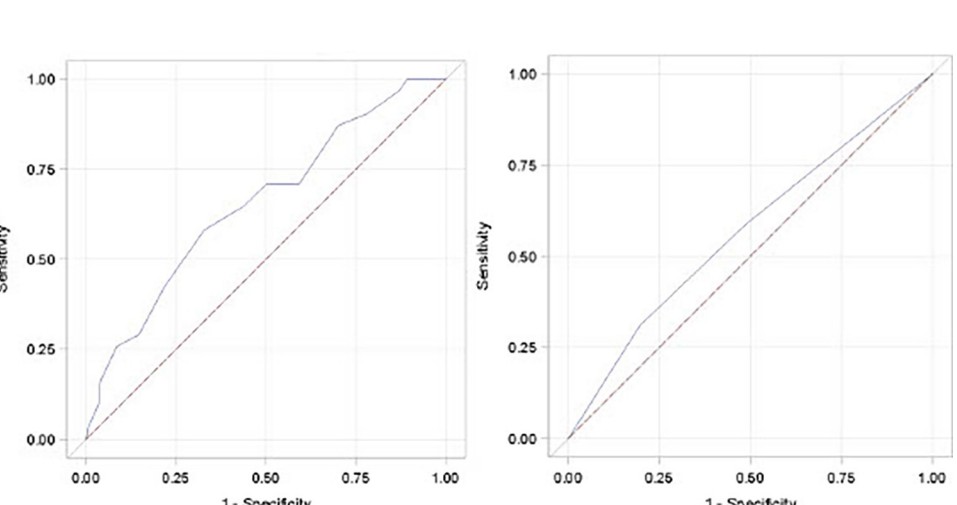

**Fig 2. Receiver Operating Curve (ROC) for LACE score and physician assessment.**

## Discussion

Our study of 21 PCPs at three sites evaluated 194 patients recently discharged from hospital. We found a moderate level of agreement between the LACE tool and physician assessment for predicting the risk of hospital readmission within 30 days after being discharged. The LACE index performed slightly better than physician assessment with moderate predictive power, although this difference was not statistically significant.

The overall rate of readmission in our study was 14.7%. The original LACE study had a readmission rate of 8% [12] which was comparable to population-based data [1]. This may be because the clinics involved in this study were mostly affiliated with tertiary care hospitals where patients may be more complex or because of secular changes over time.

A number of studies assessing the LACE tool have been performed, but these are often in certain population groups e.g. cardiovascular disease [22], chronic obstructive pulmonary disease [23] or in older people [24] and they show conflicting results. The results of this study

**Table 4. Readmission rate according to LACE and physician assessment score.**

|  | Readmission within 30 days (with respect to hospital discharge) | | |
|---|---|---|---|
|  | **No** | **Yes** | **Total** |
| **LACE risk[1]** |  |  |  |
| Low | 88 (90.7%) | 9 (9.3%) | N = 97 |
| Medium | 51 (85%) | 9 (15%) | N = 60 |
| High | 38 (74.5%) | 13 (25.5%) | N = 51 |
| **Physician risk** |  |  |  |
| Low | 89 (88.1%) | 12 (11.9%) | N = 101 |
| Medium | 50 (86.2%) | 8 (13.8%) | N = 58 |
| High | 34 (79.1%) | 9 (20.9%) | N = 43 |

[1]categorized using the LACE score (0–9 = low risk; 10–13 = medium risk; 14–19 = high risk).

compare with the Damery study [25] where increasing LACE score and certain components of the LACE index were independent predictors of readmission. A trade-off between sensitivity and specificity can be observed with increasing sensitivity and decreasing specificity as the threshold for LACE score to classify patients as high risk for 30 days readmission is decreased.

Both methods performed sub-optimally; the PCP assessment of readmission risk may not be as effective as the LACE tool. There was overlap of the 95% C.I. for both AUROC deeming the difference not statistically significant. Studies looking at clinician assessment of readmission risk have shown varied results with one demonstrating an AUC derived from ROC of 0.689 for the risk assessment completed by the discharging attending on a hospital team vs. 0.620 for the LACE tool as a predictor [26]. Allaudeen et al. found poor ability of inpatient teams (physician, nurse and case manager) to discriminate between readmissions vs. non-readmissions (AUC from ROC of 0.5786) [27]. However, PCPs have long term patient relationships that may affect their assessment differently as a result of their knowledge of a patient's social conditions and overall medical status. To the best of our knowledge there are no studies looking at the PCPs' judgment of readmission, but the results of our study are comparable to the Allaudeen study which showed poor discriminatory power of clinicians in assessing readmission risk. This may indicate that the relationship and contextual knowledge a primary care provider has with a patient does not improve assessment of readmission risk and that the information in the EMR is not readily accessible to the PCP to allow for pertinent decision making.

If the LACE score can easily be calculated and provided to the PCP by the discharging physician it may be helpful in conjunction with physician assessment in deciding who requires more intensive follow-up after discharge. Future directions may include calculating the sensitivity and specificity of a score that combines the LACE tool with physician assessment and to observe whether implementation of such a tool can reduce readmission rates.

Our study has limitations that merit emphasis. ICES data was used to collect data on emergency department use in order to calculate the LACE score. Currently in our healthcare system emergency room visits are not comprehensively reported to PCPs. As a result, it may be more practical for the hospital discharge team to provide the PCPs with a readmission risk score. The analysis was performed without taking into account the unique structure of patient-physician hierarchy (i.e. clustering) due to small sample size of this study. As there are no significant results, further adjustments for clustering would not change these results. During the study period there was a change in the process with regards to how discharge summaries were received resulting in some physician assessments being lost. Furthermore, only readmissions where discharge summaries were received by the PCP were included, and there is a possibility that some discharges were not included. This may have weakened or underestimated the sensitivity and specificity; however this would apply to both the LACE estimates as well as the PCP estimates. We did not assess the quality of the discharge summary, the instructions provided for follow-up care or if this impacted their decision making. However physician response to the survey indicates the factors most likely to influence their assessment were familiarity with the patient and the hospital diagnosis. Finally, this study did not observe effects of implementing an intervention using the LACE tool or physician risk assessment on readmission rates but could be an opportunity for future direction.

## Conclusion

The LACE index shows moderate discriminatory power in patients at increased risk for readmission after discharge, and may be superior when comparing to PCP's clinical judgement of readmission risk. Obtaining all information needed to calculate the LACE score such as

emergency department visits may be difficult to obtain in primary care. Consideration should be given to providing the LACE score in the hospital discharge summary sent to the PCP to facilitate identification and early follow up of patients at high risk for hospital readmission. Future studies can include the effectiveness of combining physician assessment with elements of the LACE tool in predicting hospital readmission.

## Supporting information

**S1 File. Re-admission risk assessment survey.**
(DOCX)

## Author Contributions

**Conceptualization:** Sakina Walji, Warren McIsaac, Rahim Moineddin, Karen Tu, Chaim M. Bell.

**Data curation:** Sumeet Kalia.

**Formal analysis:** Sakina Walji, Warren McIsaac, Rahim Moineddin, Sumeet Kalia.

**Funding acquisition:** Sakina Walji, Warren McIsaac.

**Methodology:** Sakina Walji, Warren McIsaac, Rahim Moineddin, Karen Tu, Chaim M. Bell.

**Project administration:** Sakina Walji, Michelle Levy.

**Supervision:** Sakina Walji.

**Writing – original draft:** Sakina Walji.

**Writing – review & editing:** Sakina Walji, Warren McIsaac, Rahim Moineddin, Sumeet Kalia, Michelle Levy, Karen Tu, Chaim M. Bell.

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
