## [Decision Letter · Decision Letter 0]

28 Aug 2021

PONE-D-21-06391

Predicting hospital readmission risk: A prospective observational study to compare primary care providers’ assessments with the LACE readmission risk index

PLOS ONE

Dear Dr. Walji,

Thank you for submitting your manuscript to PLOS ONE. After careful consideration, we feel that it has merit but does not fully meet PLOS ONE’s publication criteria as it currently stands. Therefore, we invite you to submit a revised version of the manuscript that addresses the points raised during the review process.

The manuscript has been evaluated by three reviewers, and their comments are available below.

All reviewers recommend greater clarity in the reporting of this manuscript, both in the Methods section and Results. Specifically, the reviewers note the need for greater detail and quantification of the primary tool used in the study, as well as additional analyses.

Could you please carefully revise the manuscript to address all comments raised?

We look forward to receiving your revised manuscript.

Kind regards,

Avanti Dey, PhD

Staff Editor

PLOS ONE

Journal Requirements:

2. Please include additional information regarding the survey or questionnaire used in the study and ensure that you have provided sufficient details that others could replicate the analyses. For instance, if you developed a questionnaire as part of this study and it is not under a copyright more restrictive than CC-BY, please include a copy, in both the original language and English, as Supporting Information.  If the original language is written in non-Latin characters, for example Amharic, Chinese, or Korean, please use a file format that ensures these characters are visible.

3. Please state whether you validated the questionnaire prior to testing on study participants. Please provide details regarding the validation group within the methods section.

5. Thank you for stating the following in the Competing Interests/Financial Disclosure* (delete as necessary) section:

“I have read the journal's policy and the authors of this manuscript have the following competing interests: Dr. Bell is a medical consultant to the Ontario Ministry of Health. The other authors have no competing interests to declare”

.We note that one or more of the authors are employed by a commercial company: Ontario Ministry of Health

a.Please provide an amended Funding Statement declaring this commercial affiliation, as well as a statement regarding the Role of Funders in your study. If the funding organization did not play a role in the study design, data collection and analysis, decision to publish, or preparation of the manuscript and only provided financial support in the form of authors' salaries and/or research materials, please review your statements relating to the author contributions, and ensure you have specifically and accurately indicated the role(s) that these authors had in your study. You can update author roles in the Author Contributions section of the online submission form.

Reviewers' comments:

Reviewer's Responses to Questions

**Comments to the Author**

1. Is the manuscript technically sound, and do the data support the conclusions?

Reviewer #1: Yes

Reviewer #2: Partly

Reviewer #3: Yes

2. Has the statistical analysis been performed appropriately and rigorously? 

Reviewer #1: Yes

Reviewer #2: N/A

Reviewer #3: No

3. Have the authors made all data underlying the findings in their manuscript fully available?

Reviewer #1: Yes

Reviewer #2: No

Reviewer #3: Yes

4. Is the manuscript presented in an intelligible fashion and written in standard English?

Reviewer #1: Yes

Reviewer #2: Yes

Reviewer #3: Yes

5. Review Comments to the Author

Reviewer #1: The article is well written and addresses one of many challenges with hospital medicine - the rate of readmission to the hospital within 30 days of discharge and how to best predict patients who are at higher risk of readmission. This would allow high risk patients to received additional attention with the goal of preventing avoidable hospital readmissions.

Abstract well structured and clear. Reflects the content of the article.

Background - Sufficiently broad and up to date, discusses limitations and utility of current prediction systems (LACE) and frames the research question clearly.

Methods - Clearly presented, ethical approval documented. Statistics are conventional and are without controversy.

Some clarity would be useful - are the patients from one hospital or for the entire province of Ontario? Some basic information like size of hospital and location (100 beds, rural area in the Province of Ontario) or if the entire province some information about the number of hospitals and the population in the service area. Also, would be useful to document if hospital admissions at different hospitals within Ontario or in other provinces would be detected.

Results - Clearly presented and well organized. Tables are clear.

Discussion - Discuss readmission detection as mentioned in the methods section - this may be a strength or a limitation for this study. Results framed in context of other knowledge about readmission prediction.

Conclusions - Not overstated or generalized. Reasonable given methods and results.

Reviewer #2: This is an interesting study comparing the use of a PCP clinical assessment of readmission risk in comparison to a previously validated tool (the LACE index). The results if this study could be used to identify pragmatic tools in defining high risk patients for readmission.

I have the following recommendations to make:

1. My biggest concern in this study is the ambiguity of the PCP assessment. The authors describe what seems to be a qualitative tool in evaluating readmission, but the analysis made was purely quantitative. I suggest further quantifying the PCP assessment tool by showing methodically how PCPs evaluated the risk. In addition, the survey used by PCP should be included in the study and described in detail. Alternatively, I suggest using qualitative research tools to better define the PCP judgment.

2. Authors are asked to clarify if patients enrolled in the study were formally evaluated in the clinic by PCP and/or performed chart check only. Also, authors are asked to provide demographic baseline data of the patient population enrolled to enhance the generalizability of the results.

3. The sensitivity and specificity analysis in the study was calculated by summing the low and medium risk groups into one group. Authors are asked to clarify the reason of doing that as it may overestimate the specificity of the results.

4. Sampling methodology in the study is unclear. Did the authors include all the charts in the registry during the study period? If not please describe how sampling was done?

5. In the discussion section the authors confusedly referencing that 31 PCPs were included in the study. Please correct to 21 PCPs and consider omitting the number of patients in the registry (as only a sample was used).

6. Authors need to report the multivariate regression results assessing the relationship between LACE and PCP assessment. Consider including more variables known to increase risk of readmission, if available (polypharmacy, types of comorbidities…etc.).

Reviewer #3: The authors of this study do a great job of comparing the LACE tool to a physicians assessment. The findings of this study have implications for everyday practice and thus, this paper stands to be highly cited in the future. I just have a few minor concerns:

First, the results section of the abstract should contain the number of patients and years of the study. This will help facilitate future systematic reviews that include this paper.

Second, the ROC AUC comparisons, should contain a p-value. Many statistical software applications, such as MedCalc and Stata, make this value readily available. Even if the difference between the AUCs are statistically insignificant, this will still help interpret the findings of the study.

Finally, it may be beneficial to clearly state in the discussion or conclusions, the implications on future practice. For example, if the authors feel that incorporating the LACE score into the electronic medical record will save (or cost) the clinicians time, this should be clearly stated in the discussion or conclusions.

Thank you for the opportunity to review this important study.

6. PLOS authors have the option to publish the peer review history of their article (what does this mean?). If published, this will include your full peer review and any attached files.

Reviewer #1: No

Reviewer #2: **Yes: **Tamer Hudali, MD, MPH, FACP

Reviewer #3: **Yes: **Joshua Parreco

---

## [Author Response · Author response to Decision Letter 0]

27 Oct 2021

Response to Academic Editor:

Response: We have adjusted the manuscript file to align with PLOS ONE’s style requirements.

2. Please include additional information regarding the survey or questionnaire used in the study and ensure that you have provided sufficient details that others could replicate the analyses. For instance, if you developed a questionnaire as part of this study and it is not under a copyright more restrictive than CC-BY, please include a copy, in both the original language and English, as Supporting Information. If the original language is written in non-Latin characters, for example Amharic, Chinese, or Korean, please use a file format that ensures these characters are visible.

Response: A copy of the survey has now been included, as Supplementary File 1. We have revised page 6 line 119 accordingly. 

3. Please state whether you validated the questionnaire prior to testing on study participants. Please provide details regarding the validation group within the methods section.

Response: As the survey was developed for this study, it has not been validated prior to use. The survey was developed by the investigators of the team to assess physician risk assessment of readmission. Response options were “low”, “medium” or high” in keeping with the LACE assessment risk categories.

4. We note that the grant information you provided in the ‘Funding Information’ and ‘Financial Disclosure’ sections do not match. When you resubmit, please ensure that you provide the correct grant numbers for the awards you received for your study in the ‘Funding Information’ section.

We have added information on the grant funding agency in the Revised Funding Statement of the revised cover letter. The funding disclosure was in relation to a competing interest on of the co-authors had, namely that they have an affiliation with the Ontario Ministry of Health. The Ontario Ministry of Health did not have any involvement in the this study and were not included in the funding information section.

5. Thank you for stating the following in the Competing Interests/Financial Disclosure* (delete as necessary) section:

“I have read the journal's policy and the authors of this manuscript have the following competing interests: Dr. Bell is a medical consultant to the Ontario Ministry of Health. The other authors have no competing interests to declare” 

We note that one or more of the authors are employed by a commercial company: Ontario Ministry of Health

Response: We agree with the editor that further clarification regarding the funding of the project is required. This study was funded by the University of Toronto’s UTOPIAN program. The commercial affiliated for Dr. Bell should only be listed as a competing interest as the Ontario Ministry of Health did not provide support for this project and did not have a role in this study. We have included a revised funding statement and a revised competing interests statement in the cover letter. 

Revised Funding Statement: This study received grant funding from the University of Toronto’s UTOPIAN Ideas to Proposal program. The funder provided support to cover the cost of conducting the study but did not have any additional role in the study design, data collection and analysis, decision to publish or preparation of the manuscript. 

Revised Competing Interests Statement: Dr. Chaim Bell, a co-author on this project, is a medical consultant to the Ontario Ministry of Health, however the Ministry of Health had no role in this study. This commercial affiliation does not alter our adherence to PLOS ONE policies on sharing data and materials. There are no additional competing interests to declare.

Response: We have revised the data availability statement to state the following. 

All data underlying the findings described in the manuscript are not fully available/without restriction. The data used within this study was obtained through the Institute for Clinical and Evaluative Sciences (ICES), and as such there are legal restrictions to sharing the dataset. As well, given that the data includes patient health information, there are ethical restrictions in regard to the sharing of data, whereby the lead study institution, Sinai Health System, would require a data transfer agreement to be approved and in place prior to transferring or sharing the data. 

Response to Reviewer 1:

1. Some clarity would be useful - are the patients from one hospital or for the entire province of Ontario? Some basic information like size of hospital and location (100 beds, rural area in the Province of Ontario) or if the entire province some information about the number of hospitals and the population in the service area. Also, would be useful to document if hospital admissions at different hospitals within Ontario or in other provinces would be detected

Response: The patients included in this study are rostered to three different primary care practices and admissions to any hospital in Ontario were captured. The total number of patients rostered to these clinics was approximately 27,500. The three different primary care sites included were in the Toronto downtown region and Greater Toronto area (all urban). 

2. Discuss readmission detection as mentioned in the methods section - this may be a strength or a limitation for this study. Results framed in context of other knowledge about readmission prediction

Response: We agree with the reviewer that the method used for readmission detection could be a limitation of the study. Only discharge summaries received by the primary care provider were captured in this study. We have included the possibility of readmission detection being underestimated due to how discharge summaries were collected as a possible limitation to the study. Please see lines 315-319 of page 15 of the manuscript. 

Response to Reviewer 2:

1. My biggest concern in this study is the ambiguity of the PCP assessment. The authors describe what seems to be a qualitative tool in evaluating readmission, but the analysis made was purely quantitative. I suggest further quantifying the PCP assessment tool by showing methodically how PCPs evaluated the risk. In addition, the survey used by PCP should be included in the study and described in detail. Alternatively, I suggest using qualitative research tools to better define the PCP judgment.

Response: In the methods section (page 6 lines 117-119) we have indicated that the PCP survey is a quantitative measure, with risk of readmission within 30 days being rated as Low, Moderate or High, by the PCP. The decision regarding the category of risk assigned to each patient’s readmission risk score was determined based on the physician’s clinical judgment, which is consistent with what would happen in clinical practice. We chose these risk categories in concordance with the LACE risk categories. To address the ambiguity regarding the PCP assessment, we have included a copy of the survey as a supplementary file. Interviews or focus groups with the primary care providers to understand their experience using the PCP survey as well as to gain a deeper insight into the PCP’s assessment could be an area for future direction.

2. Authors are asked to clarify if patients enrolled in the study were formally evaluated in the clinic by PCP and/or performed chart check only. Also, authors are asked to provide demographic baseline data of the patient population enrolled to enhance the generalizability of the results.

Response: There were no changes to usual process of care for this study, and as such patients who were included in the study were those whose discharge summary was received by the PCP. No change to usual process occurred to best mimic how the PCP assessment would be used in a real-life scenario to assess whether this assessment would be sufficient to identify patients who were high-risk for readmission and thus may benefit from more intensive follow-up post-discharge.

Table 1 (page 9) of the results section includes the demographic baseline data for the study sample. All baseline data that was made available to us has been included in that table. 

3. The sensitivity and specificity analysis in the study was calculated by summing the low and medium risk groups into one group. Authors are asked to clarify the reason of doing that as it may overestimate the specificity of the results.

Response: We agree with the reviewer that more clarity is needed regarding the methodological decision to sum the low and medium risk groups. We have including a statement in the methods section (page 7 lines 156-159) explaining that the PCP assessment’s low and medium risk categories were collapsed into a single group in order to be consistent with the LACE assessment’s categorization of high risk readmission. High risk of admission was deemed to be a >20% risk of readmission within 30 days of discharge as per the original LACE study.

4. Sampling methodology in the study is unclear. Did the authors include all the charts in the registry during the study period? If not please describe how sampling was done?

Response: All discharge summaries that were received by the three participating primary care sites during the study period were captured. Only pediatric and obstetric cases were excluded. All other discharge summaries were included.

5. In the discussion section the authors confusedly referencing that 31 PCPs were included in the study. Please correct to 21 PCPs and consider omitting the number of patients in the registry (as only a sample was used).

Response: We have edited page 13 line 262 of the discussion to show that 21 PCPS were included in the study and as suggested by the reviewer, we have omitted the total number of patients in the sampling frame.

6. Authors need to report the multivariate regression results assessing the relationship between LACE and PCP assessment. Consider including more variables known to increase risk of readmission, if available (polypharmacy, types of comorbidities…etc.).

Response: Unfortunately, the available dataset did not contain the appropriate clinical variables for us to consider in the regression framework. We acknowledge that it is necessary to control for such clinical predictors as confounders in order to generate reliable statistical inference. However, the available data tables did not allow us to achieve this task.

Response to Reviewer 3:

1. First, the results section of the abstract should contain the number of patients and years of the study. This will help facilitate future systematic reviews that include this paper.

Response: We have included the number of patients and study years within the results section of the abstract (page 2 lines 46-47).

2. Second, the ROC AUC comparisons, should contain a p-value. Many statistical software applications, such as MedCalc and Stata, make this value readily available. Even if the difference between the AUCs are statistically insignificant, this will still help interpret the findings of the study.

Response: We would like to thank the reviewers for providing this suggestion and our apologies for this oversight. We added the hypothesis test evaluation for two ROC comparisons of LACE tool and physician assessment. We updated the Methods section (page 7-8, lines 164-166) by adding the following text and reference (page 20, lines 416-418):

“We compared the AUROC of LACE tool and physician assessment tool using the non-parametric approach, as further described by DeLong, Delong and Clarke-Pearson (1988).20”

Reference:

20. DeLong ER, DeLong DM, Clarke-Pearson DL. Comparing the Areas under Two or More Correlated Receiver Operating Characteristic Curves: A Nonparametric Approach. Biometrics. 1988 Sept; 44(3):837–845

We determined the difference between the two ROC metrics as statistically insignificant (evaluated at nominal coverage rate of 5%). The p-value was computed to be 0.08 and this is also described in the results section of the manuscript (page 11-12, lines 229-232):

“However, the AUROC of LACE readmission was not statistically significant at the nominal level of 5% to the AUROC of physician assessment (Chi-square= 3.11, P-value= 0.08).”

3. Finally, it may be beneficial to clearly state in the discussion or conclusions, the implications on future practice. For example, if the authors feel that incorporating the LACE score into the electronic medical record will save (or cost) the clinicians time, this should be clearly stated in the discussion or conclusions.

Response: In the conclusion section (page 15 lines 312-317), we have discussed future considerations in regard to these tools.

---

## [Editor Report · Decision Letter 1]

22 Nov 2021

Predicting hospital readmission risk: A prospective observational study to compare primary care providers’ assessments with the LACE readmission risk index

PONE-D-21-06391R1

Dear Dr. Walji,

We’re pleased to inform you that your manuscript has been judged scientifically suitable for publication and will be formally accepted for publication once it meets all outstanding technical requirements.

Kind regards,

Tamer Hudali

Guest Editor

PLOS ONE

Additional Editor Comments (optional):

I participated as a reviewer for the initial evaluation of this manuscript.
---

## [Editor Report · Acceptance letter]

2 Dec 2021

PONE-D-21-06391R1 

Predicting hospital readmission risk: A prospective observational study to compare primary care providers’ assessments with the LACE readmission risk index 

Dear Dr. Walji:

I'm pleased to inform you that your manuscript has been deemed suitable for publication in PLOS ONE. Congratulations! Your manuscript is now with our production department. 

Kind regards, 

on behalf of

Dr. Tamer Hudali 

Guest Editor

PLOS ONE